# Targeting *TET2* as a Therapeutic Approach for Angioimmunoblastic T Cell Lymphoma

**DOI:** 10.3390/cancers14225699

**Published:** 2022-11-20

**Authors:** Lina Hu, Xuanye Zhang, Huifeng Li, Suxia Lin, Shengbing Zang

**Affiliations:** 1Sun Yat-sen University Cancer Center, State Key Laboratory of Oncology in South China, Collaborative Innovation Center for Cancer Medicine, Guangzhou 510060, China; 2Department of Pathology, Sun Yat-sen University Cancer Center, Guangzhou 510060, China; 3Department of Medical Oncology, Sun Yat-sen University Cancer Center, Guangzhou, China

**Keywords:** AITL, *TET2*, 5hm, DNA methylation, DNA demethylation, *RHOA*, *IDH2*, *DNMT3*, drug target

## Abstract

**Simple Summary:**

Recent advances in exome sequencing and comprehensive genomic studies have shed new light on the pathogenesis and mutational landscape of Angioimmunoblastic T-cell lymphoma (AITL). The essential pathogenic role of recurrent *TET2*, *DNMT3A*, *IDH2,* and *RHOA* mutations in AITL has been identified. Notably, the detection of *TET2* mutations in bystander B cells in AITL samples has allowed new mechanistic advances in AITL combined with B-cell lymphoma. In this review, we highlight the current role of *TET2* mutations in the pathogenesis of AITL and how mutational crosstalk in *TET2* and other partner genes (*RHOA*, *DNMT3A,* and *IDH2*) cooperate to contribute to the disease. In addition, we elaborate on recent advances in AITL frequently comprising B-cell lymphoma and discuss the potential prospects of targeting *TET2* as an anti-AITL agent.

**Abstract:**

Angioimmunoblastic T-cell lymphoma (AITL), a type of malignant lymphoma with unique genomic aberrations, significant clinicopathological features, and poor prognosis, is characterized by immune system dysregulation. Recent sequencing studies have identified recurrent mutations and interactions in tet methylcytosine dioxygenase 2 (*TET2*), ras homology family member A (*RHOA*), DNA methyltransferase 3 alpha (*DNMT3A*), and mitochondrial isocitrate dehydrogenase II (*IDH2*). Notably, since B-cell lymphomas are frequently observed along with AITL, this review first summarizes its controversial mechanisms based on traditional and recent views. Epigenetic regulation represented by *TET2* plays an increasingly important role in understanding the multi-step and multi-lineage tumorigenesis of AITL, providing new research directions and treatment strategies for patients with AITL. Here, we review the latest advances in our understanding of AITL and highlight relevant issues that have yet to be addressed in clinical practice.

## 1. Introduction

Angioimmunoblastic T-cell lymphoma (AITL) is a malignant hematologic tumor derived from T follicular helper (TFH) cells. AITL is among the most aggressive of non-Hodgkin lymphomas with a poor prognosis [1]. Only a small proportion of AITLs are detected at an early stage, which is amenable to the current treatment. Nonetheless, AITLs are mostly diagnosed at an advanced stage, and patients with advanced AITLs show a limited response to intensified chemotherapy along with high relapse rates, mainly due to a lack of effective targeted therapies. Therefore, an in-depth understanding of the pathogenesis of AITL will provide a mechanistic basis for the design of effective therapeutic regimens.

Recent exome sequencing and comprehensive genomic studies of AITL samples have led to the discovery of frequent mutations in three genes (*TET2*, *DNMT3A,* and *IDH2*) that are directly or indirectly involved in the regulation of DNA methylation or hydroxymethylation [2,3,4]. Mutations in *TET2,* which is a member of the ten-eleven translocation (TET) family of 2-oxoglutarate (2OG)-dependent dioxygenases, were more frequent than the other mutations. Moreover, *TET2* mutations frequently coexist with *RHOA* or *DNMT3A* in AITL [5,6,7]. Loss of *TET2* function leads to the defect in successively oxidizing 5-methylcytosine (5mC) to 5-hydroxymethylcytosine (5hmC), 5-formylcytosine (5fC), and 5-carboxylcytosine (5caC) in the mammalian genome [4]. Thus, *TET2* plays a crucial role in the pathogenesis of AITL through active cytosine demethylation.

In this review, we focus on exploring the current role of *TET2* mutations in the pathogenesis of AITL and how mutations in *TET2* as well as other genes (*RHOA*, *DNMT3A*, and *IDH2*) cooperatively contribute to disease pathogenesis. Recently, *TET2* mutations have been detected in the AITL tumor clones of bystander B cells in AITL samples [8,9,10]. Therefore, it is likely that genetic lesions of *TET2* are critical in the co-occurrence of T and B-cell lymphomas in the same patient. Here, we elaborate a summary of *TET2* mutations in the frequent co-occurrence of AITL and B-cell lymphomas (e.g., diffuse large B-cell lymphoma and classic Hodgkin lymphoma) and discuss the potential of targeting *TET2* as an anti-AITL agent.

## 2. AITL and Other Lymphomas with a TFH-Cell Phenotype

According to a review of the upcoming fifth edition of the World Health Organization Classification of Haematolymphoid Tumors update on mature T-and natural killer (NK) cell neoplasms, AITL and other lymphomas with a TFH-cell phenotype have been regarded as entities in the family of nodal T-follicular helper cell lymphomas (nTFHLs) [11]. This family includes three entities of nodal T-cell lymphomas that arise from TFH cells, which are defined by the expression of at least two (ideally three) TFH markers: CD10, BCL-6, PD1, CXCL13, and ICOS. Diseases previously referred to as “angioimmunoblastic T-cell lymphoma” “follicular T-cell lymphoma” and “peripheral T-cell lymphoma with TFH phenotype” have been renamed as nTFHL angioimmunoblastic-type (nTFHL-AI), nTFHL follicular-type (nTFHL-F) and nTFHL not otherwise specified (nTFHL-NOS), respectively. Cutaneous T-cell lymphomas with a TFH-cell phenotype were excluded. 

The prototype of these lymphomas is nTFHL-AI (referred to as AITL in this paper), which has distinct clinical and histological features. AITL is typically manifested by lymphadenopathy, hepatosplenomegaly, polyclonal hyperglobulinemia, and systemic symptoms. Skin rashes are frequently observed, as is pruritus. Other common findings are pleural effusion, arthritis, and ascites [12,13]. The histology of AITL is very unique: (1) AITL tumor cells which are small to medium in size typically have abundant pale cytoplasms with round and slightly irregular nuclei; (2) AITL tumor cells show the cell-of-origin of TFH cells with expression of five TFHcell-associated genes (CD10, BCL-6, PD1, CXCL13, and ICOS) in most cases. (3) A polymorphous inflammatory background that contains variable numbers of reactive lymphocytes, histiocytes, plasma cells, and eosinophils; (4) Arborization of high endothelial venules (HEVs) and follicular dendritic cells (FDCs) are very prominent; (5) Variable numbers of B immunoblasts are usually present in the paracortex and may be positive or negative for EBV. More recently, studies based on next-generation sequencing have determined frequent mutations in genes encoding epigenetic modifiers, such as *TET2*, *DNMT3A*, and *IDH2*, as well as the small GTPase *RHOA*. Among these, *IDH2 R172* mutation apparently has specificity for AITL [2,7,8], while others are visible in other peripheral T-cell lymphomas, particularly those with a TFH-cell phenotype. This suggests a consistent oncogenic course in these related types of lymphomas.

nTFHL follicular-type (nTFHL-F) is a rare subtype of nTFHLs, with a predominantly follicular growth pattern that distinguish from AITL in the histological features [10]. Patients with nTFHL-F resemble AITL, characterized by lymphadenopathy, hepatosplenomegaly, systemic symptoms, polyclonal hypergammaglobulinemia, and autoimmune manifestations. nTFHL-F and AITL have very similar genetic profiles with mutations in *TET2*, *DNMT3A*, and *IDH2*, as well as the small GTPase *RHOA* [14,15]; however, nTFHL-F has specific genetic alterations due to the translocation t (5;9) (q32; q22) leading to *ITK*-*SYK* fusion, which is rarely seen in AITL [16,17]. 

nTFHL-NOS, previously classified as peripheral T-cell lymphomas, not otherwise specified (PTCL-NOS), is now included in the family of nodal T-follicular helper cell lymphomas. This lymphoma lacks the characteristic histological features (e.g., inflammatory infiltration, vascular hyperplasia, or dilation of follicular dendritic cell meshwork) of AITL but presents with a TFH-cell phenotype and genetic abnormalities similar to AITL. 

## 3. The Highly Recurrent *TET2* Mutation in AITL

TET proteins were named due to the rare “ten-eleven translocation” and included TET1, TET2, and TET3 [4]. TET1 and TET3 are virtually mutation-free in hematological malignancies. However, the gene located in the 4q24 region of human chromosome 4, which encodes *TET2,* frequently undergoes microdeletions and copy number-neutral loss-of-heterozygosity in various hematological malignancies [4,18]. Most missense mutations in *TET2* impair enzyme activity and induce aberrant DNA methylation [4,18].

More recently, *TET2* mutations have been frequently observed in T-cell lymphomas, particularly in AITL and other lymphomas with a TFH-cell phenotype. Table 1 summarizes the studies that have included TFH cell lymphoma and PTCL-NOS. The techniques used for the detection of *TET2* mutations are listed in Table 1. According to previous studies, *TET2* mutations have been detected in 47–100% of AITL cases [8,18,19,20,21,22,23,24,25,26,27]. Among patients with *TET2* mutations, over 57% harbored two or more mutations. Mutations in *TET2* were identified through the entire coding region, while most of them were insertions/deletions generating frameshift and nonsense mutations. Only one study assessed *TET2* mutations in nTFHL-F, with a mutation rate of 75% [10]. Up to 58–73% of nTFHL-NOS had *TET2* mutations [10,18,20]. The mutation rate of *TET2* in PTCL-NOS ranged from 14.6% to 38% [10,18,20]. In conclusion, *TET2* has the highest mutation rate in AITL (up to 100%), which is more common than that in other tumors.

In AITL, *TET2* mutations are associated with advanced-stage disease [18,22], thrombocytopenia [18], high International Prognostic Index (IPI) scores [18,28], an increased number of involved extranodal sites [18], the presence of B symptoms [22], and elevated lactate dehydrogenase (LDH) levels [22]. Elevated LDH levels and positive B symptoms are considered indicators of tumor load and play an overwhelming role in tumor maintenance [29], thereby suggesting that *TET2* mutations can lead to increased tumor burden.

Whether *TET2* mutations affect the prognosis of AITL patients is unclear. Two studies have shown that *TET2* mutations are correlated with shorter progression-free survival (PFS) [6,18], whereas another study found no significant difference in OS or PFS when compared between the group of wild-type *TET2* and the mutant group; the only individual mutation that significantly reduced survival was *IDH2* [22]. *TET2* and *IDH2* co-occurrence possessed better PFS than *TET2* mutations alone, which also demonstrated the emphasis on epigenetic modifications [22]. In conclusion, the prognostic value of *TET2* mutations in AITL remains controversial and is still being explored. 

## 4. Crosstalk between *TET2* and Other Genes in AITL

### 4.1. Interaction between TET2 and RHOA Mutations

Ras homology family member A (*RHOA*) functions as a small GTPase, which is activated by a specific guanine nucleotide exchange factor (GEF), to promote the exchange of GTP and GDP [33,34]. Missense mutations in the pG17V substitution encoding the RHOA GTPase were detected in 50–70% of AITL patients [6,7,35,36]. As a result, *RHOA G17V* loses its GTP-binding activity to inhibit RHO signaling by isolating the GEF protein [6,7,37]. Thus, *RHOA* mutants promote the development of PTCL by dominantly and negatively inhibiting the binding of wild-type RHOA protein to GTP [6,7]. Notably, the *RHOA* mutation is specific to T-cell lymphoma and absent in B-cell lymphoma [37].

*VAV1* (Vav Guanine Nucleotide Exchange Factor 1) has been characterized as a *G17V RHOA*-specific binding companion [38]. This protein not only acts as a GEF (Guanine Nucleotide Exchange Factor) to activate small GTPases in the T-cell receptor (TCR) signaling pathway but also acts as an adapter to promote the formation or function of the TCR signaling complex [38,39]. The *G17V RHOA*-*VAV1* axis activated in AITL accelerates TCR signal transduction by enhancing phosphorylation at 174Tyr [38,40]. Genes with the highest mutation frequency in the TCR signaling pathway include *PLCG1*, *CD28*, *PIK3*, *GTF2I,* and *CTNNB1* [35], which are related to cell activation and TFH-derived lymphoma [35]. In addition, *RHOA G17V* expression in CD4+ T cells promotes Tfh lineage specification and AITL transformation through a mechanism dependent on the *ICOS*-*PI3K*-*mTOR* signaling pathway [40,41]. The blocking of the *ICOS*-*PI3K*-*mTOR* signaling pathway provides an innovative and possible strategy for targeting *RHOA G17V* therapy for AITL. As a costimulatory molecule and migration receptor for Tfh cells, *ICOS* co-receptor signaling phosphorylates transcription factor *FOXO1* (forkhead box O1) and decreases nuclear transcription factor KIF2 [42,43], which is essential for driving Tfh lineage differentiation and maintaining the Tfh phenotype. Therefore, *RHOA G17V* induces Tfh lineage differentiation and drives AITL transformation by accelerating TCR signal transduction and enhancing the ICOS-*PI3K* signaling pathway.

Notably, *TET2* mutation and *RHOA G17V* have a synergistic effect on human AITL pathogenesis, with co-mutations of *TET2* and *RHOA* observed in 60–70% of AITL cases [6,7,36], leading to AITL-like disease with high penetrance [41,43,44]. In multiple mouse models reported, *Tet2* loss and *RHOA G17V* expression jointly led to AITL, which supports a multi-step model of AITL development (Figure 1) [41,43,44,45]. *TET2* deletion with *RHOA G17V* expression can lead to abnormal activation of CD4+ T cells and the imbalance of T cell homeostasis in the peripheral blood, which results in a reduction of Treg cells and an increase of Tfh cells [43]. In addition, the loss of function of *RHOA* and *TET2* together interfere with *FOXO1* function, which is manifested by inhibiting *FOXO1* transcription and promoting its phosphorylation, thereby inducing the differentiation of Tfh cells and greatly reducing the need for *ICOS* signal transduction [42,43]. In most cases, *RHOA* mutations are specifically identified only in tumor cells, whereas *TET2* mutations are found in both tumor and non-tumor hematopoietic cells [44]. The allele frequencies of *TET2* were distinctly higher when compared to the frequencies of *RHOA* and *IDH2*. *DNMT3A* and *TET2* mutations have a large degree of overlap and similar allele burden, whereas *RHOA* and *IDH2* mutations share similar allele frequencies [7]. This indicates that *TET2* and/or *DNMT3A* mutations precede *RHOA* and/or *IDH2* mutations in the majority of cases [7], while *RHOA G17V* is the secondary strike of AITL pathogenic mechanism relay *TET2* in early hematopoietic progenitor cell mutation.

### 4.2. Interaction between TET2 and DNMT3A Mutation

In AITL, *DNMT3A* mutations occur at a frequency of 20–38.5% [6,7,24,30]. The majority of *DNMT3A* mutations are concentrated in the MTase-binding domain, which can lead to loss-of-function. Notably, Arg-882 (R882), located in the MTase binding domain, was the most common mutation site [24]. *DNMT3A* deficiency progressively impairs hematopoietic stem cell (HSC) differentiation while amplifying the number of HSCs in the bone marrow [46], thereby, making hematopoietic stem cells prone to malignant transformation [47].

*DNMT3A* is a de novo DNA methyltransferase that mediates DNA methylation [48], whereas the TET protein family can oxidize 5-methylcytosine (5-mc) to 5-hydroxymethylcytosine (5-hmc) [49] during DNA demethylation. Thus, the interaction between *TET2* and *DNMT3A* co-regulates the addition or removal of DNA methyl groups [48,49]. The *DNMT3A R882H* mutant combined with *TET2* inactivation plays a role in DNA methylation dysregulation and induces malignant tumors in the mouse lymphatic system [5].

Interestingly, *TET2* and *DNMT3A* mutations co-occur significantly, particularly in AITL [3,5,6,7,30]. *DNMT3A* and *TET2* mutations have a large degree of overlap and similar allele burden. These mutations have been found not only in tumor cells from patients with T-cell lymphoma [3,4,5,8,30,48,50] but also in multiple lineages of normal bone marrow and blood cells [3,4,7,51,52], including B cells, myeloid cells, and myeloid progenitor cells [7,8], whereas *RHOA* and *IDH2* mutations are confined only to tumor cells [7,8]. These data provide further confirmation of the hypothesis that *TET2* and *DNMT3A* mutations may exist in precancerous cells.

### 4.3. Interaction between TET2 and IDH2 Mutations

Mitochondrial isocitrate dehydrogenase II (*IDH2*) is an NADP+-dependent enzyme that is critical for cell proliferation and catalyzes the conversion of isocitrate to α-ketoglutarate (αKG) in the mitochondria [53]. However, the *IDH2 R172* mutant lacks this ability and catalyzes the NADPH-dependent reduction of αKG to R(−)-2-hydroxyglutarate(2HG) [53,54,55], contributing to the malignant transformation [54,55]. Since 2HG is a structural analog of αKG, mutant *IDH*-induced increases in 2HG levels may interfere with the normal cycle of DNA methylation and demethylation by inhibiting the activity of α-KG-dependent dioxygenases, such as TET family proteins [55,56]. *IDH2 R172* mutant-mediated DNA hypermethylation inhibits T-cell receptor signal transduction and T-cell differentiation, thereby supporting the idea that 2HG produced by the mutant enzyme is responsible for driving tumorigenesis and progression [24,57].

In acute myeloid leukemia (AML), the *IDH1/2* and *TET2* mutations repel each other. *IDH1/2* mutations mediate hypermethylation and suppress *TET2*-mediated demethylation, impelling myeloid differentiation [55]. *IDH2* mutations are seemingly specific events in AITL, with a mutation frequency of 20–45% [2,7,8,24,30,58], most of which coexist with *TET2* mutations [7,24,30,58]. It is interesting to note that the *IDH2* mutation in AITL is hypothesized to constitute a second strike and may refine the differentiation of precancerous clones [7,19]. Cases with *IDH2* and *TET2* double mutations showed the upregulation of TFH-related genes (*IL21* and *ICOS*) and an increased TFH cell-like phenotype, thereby defining a distinct subgroup of AITL with unique follicular T helper gene expression signatures [24].

Therefore, *IDH2 R172* mutations can not only affect epigenetic regulation by competitively inhibiting *TET2* demethylation but also enhance the expression profile of TFH and promote AITL development.

## 5. Role of *TET2* Mutations in B-Cell Lymphoma Observed in AITL

In clinical practice, the incidence of B-cell lymphoma in the setting of AITL is as high as 10% [59,60,61,62]. However, the reason for this unique-recurring, frequent concomitant appearance of B- and T-cell lymphomas is not known.

### 5.1. Traditional View: AITL Precedes B-Cell Lymphoma

B-cell lymphoma has traditionally been regarded as a follow-up to AITL, as indicated by this model (Figure 2). Latent Epstein-Barr virus (EBV) infection is a common feature of sporadic non-neoplastic bystander B cells, showing that B-cells are usually positive for EBV-coding RNA in 66–97% of companion/secondary B-cell lymphomas [10,59,60,61,62,63]. EBV has a significant effect on B cell differentiation, leading to the survival and clonal expansion of Ig-less B cells, and promoting the progression of B-cell lymphoma [59,60,62,64]. Zhou et al. suggested that the reduction in antiviral immune surveillance in AITL may be conducive to the proliferation of potential EBV-infected B lymphocytes and the subsequent progression of EBV+ B cell malignancies [59]. Furthermore, the B cell-stimulating properties of TFH tumor cells may promote B cell expansion [9,65]. Although EBV infection may contribute to the oncogenic mechanism of EBV-positive B-cell lymphoma, a considerable proportion of patients with B-cell lymphoma are EBV-negative [60,66,67], suggesting that other factors besides EBV may be responsible for the occurrence of B-cell lymphoma.

### 5.2. Recent View: AITL Occurs Concurrently with B Cell Lymphoma

*TET2* mutations represent early genetic lesions that occur in upstream hematopoietic stem/progenitor cells (HSC/HPC) and have been detected in both tumor tissue and normal blood cells of patients with AITL [3,4,7,30]. Between 10% and 60% of polyclonal B cells in AITL lymph nodes carry the same *TET2* mutation in their corresponding T-cell lymphoma clones [9]. Furthermore, premalignant cells with *TET2* mutations may differentiate into not only T-lineage tumor cells but also B cells [8,9]. Single-cell analysis revealed that 7–66% of B cells were derived from the differentiation of *TET2*-mutated HSC rather than through clonal expansion of a minority of B cells with *TET2* mutations [9,64]. Notably, all *NOTCH1* mutations in B-cell lymphomas secondary to AITL were detected only in B cells, whereas *RHOA G17V* mutations were specific to all tumor T cells [8], showing that *NOTCH1* mutation and *RHOA G17V* mutation play a role in B-cell lymphoma and AITL, respectively, and both are secondary mutations following *TET2* mutation. Consequently, AITL with B-cell lymphoma is a multistep alteration, with AITL and B-cell lymphoma occurring in the T-cell zone and germinal center, following the differentiation of hematopoietic stem cells carrying the *TET2* mutation into premalignant T-cell and B-cell lineages, respectively. 

Taken together, these studies suggest that clonal hematopoietic cells carrying the *TET2* mutation may acquire the *NOTCH1* mutation when they differentiate into B cells, which may subsequently enhance the effects of the *TET2* mutation. The co-occurrence of AITL and B-cell lymphoma was mediated by B cells harboring *TET2* and *NOTCH1* mutations, and Tfh cells carrying *TET2* and *RHOA* mutations, as indicated by the model outlined in Figure 3.

## 6. New Drugs Targeting *TET2* for Treatment of AITL

Most peripheral T-cell lymphomas (PTCLs) have a high degree of malignancy, rapid disease progression, poor clinical prognosis, low remission rates, and high recurrence rates after first-line treatment. Despite several novel drugs currently being investigated [68], there has been little improvement in the prognosis of AITL patients over the past 20 years [69]. According to the 2022 NCCN guidelines, anthracycline-based chemotherapy regimens [e.g., CHOP (cyclophosphamide, doxorubicin, vincristine, and prednisone), CHOP + etoposide (CHOEP), or dose-adjusted EPOCH (etoposide, prednisone, vincristine, cyclophosphamide, and doxorubicin)] are the most commonly used first-line therapy regimens, although with a high failure rate and frequent relapse in PTCL. Additionally, epigenetic drugs have been applied in clinical trials (Table 2) and recommended as second-line therapy by NCCN clinical practice guidelines [70], providing new research directions to promote promising clinical outcomes as a focus of research in PTCL [71]. 

### 6.1. Histone Deacetylase Inhibitor (HDACis) 

Histone deacetylase (HDAC) is an enzyme engaged in chromatin remodeling and regulates the epigenetics of gene expression [82]. HDAC inhibitors, such as romidepsin and belinostat, have previously been demonstrated to induce acetylation of histones and other proteins [83,84], producing antitumor activity by inhibiting gene transcription, regulating cell cycle, and increasing tumor cell apoptosis [83,85].

A pivotal, open-label phase II clinical trial of romidepsin for relapsed or refractory PTCL involving 130 patients with an objective response rate (ORR) of 25% and a complete response/unconfirmed complete response rate (CR/CRu) of 15% demonstrated that single-agent romidepsin induced complete and durable responses with manageable toxicity [72]. These results contributed to the FDA’s (Food and Drug Administration) approval of romidepsin in June 2011. Since then, several pivotal multicenter phase II studies have shown the activity of HDAC inhibitors as monotherapy in patients with relapsed or refractory PTCL (ALCL, ALK-negative, PCTL-NOS, and AITL) [73,74]. The combination of romidepsin and conventional CHOP chemotherapy did not seem to have the desired effect. A non-randomized phase 1b/2 study of 37 previously untreated patients with PTCL showed that the combination of romidepsin was more toxic than expected with CHOP alone (panel) [77]. Another homogeneous phase 3 combination trial involving 421 previously untreated PTCL patients also showed that the combination did not improve OS, PFS, or response rates, but instead increased the frequency of grade ≥3 therapeutic adverse events [79]. In contrast, the combination of romidepsin, duvelisib (a PI3K-δ,γ Inhibitor), and 5-azacytidine showed good efficacy in clinical trials of T-cell lymphoma, with the highest ORR and CR of 61% and 48%, respectively [41,78,80].

Chidamide is a novel selective HDAC inhibitor characterized by a unique mechanism of action. Two chidamide monotherapies and two combination therapies were administered in PTCL [75,76,81]. Among them, combination therapy has shown more promising results than monotherapy. The ORR and CR of chidamide combined with 5-azacytidine were 68.8% and 31.2%, respectively, which had the best results. Panobinostat is a potent oral pan-deacetylase inhibitor as well. In a clinical trial, the combination of panobinostat and the proteasome inhibitor bortezomib was shown to be effective in patients with relapsed or refractory peripheral T-cell lymphoma [76].

### 6.2. Hypomethylating Agents (HMAs)

Hypomethylating agents (HMAs), for instance 5-azacytidine (Aza) and decitabine, can balance the hypermethylated state of DNA induced by mutations in epigenetic regulators such as *TET2*, DNMT3A, and *IDH2*. HMAs have been authorized for the treatment of myelodysplastic syndromes (MDS) in many countries, and MDS patients with *TET2* mutations show an increased response rate to HMAs [86].

One study showed that *TET2* mutations are associated with a better response to azacytidine and revealed that the *TET2* gene is the strongest biomarker of clinical response [87]. Since *TET2* mutations are occurring more frequently in AITL than in MDS, the therapeutic effect of Aza was expected to be more in the case of AITL. In a retrospective study, Aza et al. showed promising results, where out of the 12 AITL patients with *TET2* mutations, nine responded to Aza, including six complete responses (CRs) and three partial responses (PRs), resulting in an ORR of 75% [31]. Two case reports showed complete remission in AITL patients with *TET2* mutations after the administration of Aza [28,88], suggesting that targeting aberrant DNA methylation in AITL may be as a potential strategy to alternate previous chemotherapy regimens. Furthermore, in a phase I clinical trial investigating the efficacy of Aza combined with romidepsin in 33 patients with PTCL, 10 (32%) obtained an objective response, and seven (23%) achieved a complete response. These results indicate that the combination of epigenetic modification drugs exhibits promising therapeutic activity in patients with PTCL [89].

Decitabine (5-aza-2’deoxycytidine) is a more effective methyltransferase inhibitor than 5-azacytidine (Aza), and its combination with HDAC inhibitors has a synergistic effect on reactivating gene expression [90]. Decitabine has been demonstrated to have potent antineoplastic activity in anaplastic cell lymphoma (ALCL) and MDS [91,92]. A study showed that the combination of the PARPi niraparib (Npb), HDACi romidepsin (Rom), and HMA decitabine (DAC) or panobinostat (Pano) was synergistically cytotoxic to leukemia and lymphoma cells and inhibited cell proliferation by up to 70% through activating the ATM pathway [93]. To date, no study has combined AITL therapy with decitabine. 

### 6.3. Preclinical Trials of TET2 Targeting Agents 

DNA methylation changes resulting from the silencing or activation of genes by *TET2* mutations play a key role in AITL pathogenesis. At present, there are no clinically approved drugs capable of reactivating TET function; however, many preclinical trials have been conducted on *TET2*-related drugs (Table 3), thereby representing potential therapeutic opportunities.

#### 6.3.1. TET Enzyme Inhibitors

Recently, novel cytosine-based TETase suppressors have been identified. A promising cytosine-based lead compound, Bobcat339, reduced the abundance of DNA 5hmC by inhibiting TETase activity in cells [94]. Furthermore, a TET-selective small-molecule inhibitor (TETi76) was effective in limiting the clonal growth of *TET2* mutants and reduced cellular hydroxymethylation both in vitro and in vivo [93]. However, the limitations of these *TET2* inhibitors restrict their clinical application. Bobcat339 has not been tested in animal models and clinical trials, and its effects and toxicities among humans are unknown. Furthermore, while TETi76 decreases cytosine hydroxymethylation levels and limits clonal growth of *TET2* mutant cells in vitro and in vivo, a potential drawback of TETi is that it may replicate the *TET2* mutation and drive the malignant transformation of normal hematopoietic cells. More validation and discussion are needed for their future application.

#### 6.3.2. TET Enzyme Cofactors

Recent studies on embryonic stem cells have shown that ascorbic acid (AA)/Vitamin C is a cofactor for TET. Ascorbic acid is capable of binding to the catalytic domain of TET to promote the oxidation of 5-methylcytosine and DNA demethylation [96,103,104,105]. AA can exert beneficial anti-proliferative effects on AML cells carrying co-existing *TET2* and *TP53* mutations without interfering with targeted cytotoxic therapies [97]. Additionally, the combination of AA and a class I/II histone deacetylase inhibitor (HDACis) can prevent *TET2* mutant myeloid neoplasia (MN), thereby providing a preclinical theoretical basis for further research [98]. Furthermore, AA can combine with 5-AZA to further upregulate methylated genes and human endogenous retroviruses (HERVs) [99]. In addition, vitamin C also promotes decitabine- or azacitidine-induced DNA hydroxymethylation and reactivation of the tumor suppressor *CDKN1A* [100]. The dual use of AA and vitamin C may improve the response to epigenetic therapy. 

The limited early studies suggest that AA levels may vary in different tumor patients, suggesting the need to clarify who to administer, the route of administration, and the dose. Therefore, prospective studies are urgently required to elucidate the incidence of AA deficiency according to tumor type and stage, and whether blood levels of AA correlate with the methylation landscape of tumors to maximize patient benefit. 

#### 6.3.3. Other Possibilities

Transcriptomic profiling showed that a four antibiotic cocktail therapy inhibited the differentiation and proliferation of *TET2*-loss myeloid and lymphoid tumor cells in vivo by mediating genetic alterations in the TNF-α signaling pathway [101]. However, how TNF-α signaling is activated and how antibiotic treatment inhibits pathways associated with TNF remain unresolved. Ginkgo biloba extract can also inhibit cell expansion and invasion by increasing *TET2* expression in colorectal cancer cells through miR-29a [102]. These findings suggest that new molecular biology tools will provide effective insights into the field of epigenetics and that TET inhibitors may prove to be promising targeted drugs in *TET2* mutational tumorigenesis.

## 7. Conclusions

In this review, we have summarized the synergistic effects of *TET2* mutations and additional genes (*RHOA*, *DNMT3A*, and *IDH2*) in the pathogenesis of AITL. Evidence of “multi-step”and“multi-lineage” genetic events in AITL can provide insights and ideas for the origin and occurrence of this unique subtype of T-cell lymphoma. Identification of novel therapeutic vulnerabilities has induced a significant shift in the clinical management of AITL. The success of epigenetic drugs for AITL, particularly HDAC inhibitors and hypomethylation drugs, offers new hope for patients with refractory AITL. Continued progress in preclinical experiments based on cell lines and animal models will help improve the survival rate of patients with *TET2* mutant malignant tumors. Furthermore, challenges and opportunities coexist, and many opportunities for improvement remain, including the identification of biomarkers for targeted treatment response, the description of drug resistance mechanisms, and the development of successful AITL combination therapies. 

## Figures and Tables

**Figure 1 cancers-14-05699-f001:**
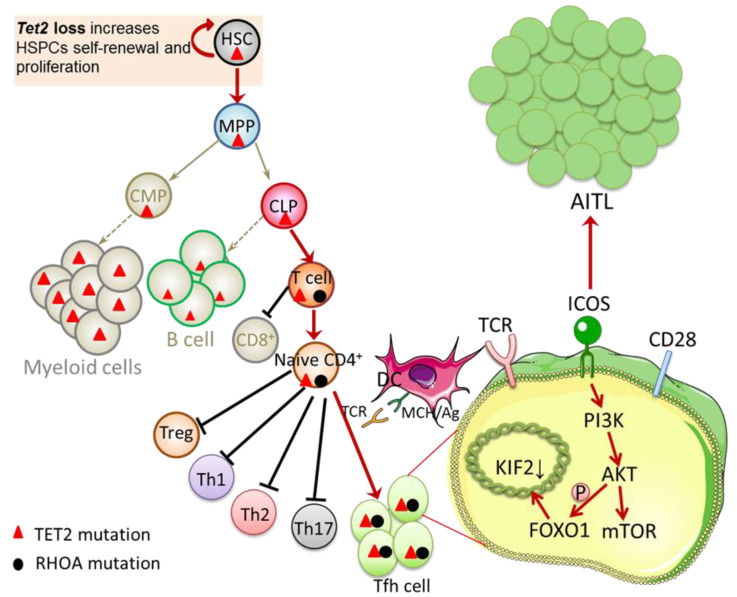
Interaction between *TET2* and *RHOA* mutations. *TET2* loss increases HSPCs self-renewal and proliferation. Cooperativity between *TET2* loss and *RHOA G17V* in T cell lineage suppresses the CD8+ T cell differentiation and endows the naïve CD4+ T cells with a competitive advantage, skews cells differentiation toward Tfh cells, and induces abnormal Tfh cell activation and transformation. *ICOS* exerts its costimulatory function via the *ICOS*-*PI3K*-*mTOR* signaling pathway, which is essential in driving Tfh lineage differentiation and maintaining the Tfh phenotype. In addition, *ICOS* activates *AKT* via *PI3K*. By phosphorylating the transcription factor *FOXO1*, *AKT* reduces the transcription factor KIF2, leading to a significant increase in the number of Tfh cells. Both of these pathways play an essential role in promoting Tfh lineage specification and AITL transformation. HSC, hematopoietic stem cells; MPP, multipotent blood progenitors; CLP, common lymphoid progenitors; CMP, common myeloid progenitors; Treg, regulatory T cell; Th1, T helper 1 cell; Th2, T helper 2 cell; Th17, T helper 17 cell; Tfh cell, follicular helper T cell; DC, dendritic cell; MHC/Ag, antigen presented on major histocompatibility complex; TCR, T-cell receptor; *ICOS*, inducible T-cell co-stimulator; *PI3K*, phosphatidylinositol-3-kinase; *AKT*, protein kinase B; *mTOR*, mammalian target of rapamycin; *FOXO1*, forkhead box O1; AITL, angioimmunoblastic T-cell lymphoma.

**Figure 2 cancers-14-05699-f002:**
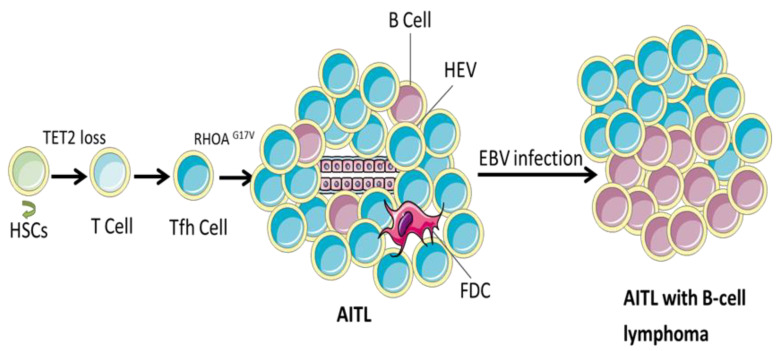
Traditional view: AITL precedes B-cell lymphoma. Early and subsequent oncogenic events, such as *TET2* loss and the acquisition of *RHOA G17V* synergize to induce the cause of AITL. On this basis, EBV infection of sporadic non-neoplastic bystander B cells has a significant impact on the differentiation of B cells, allowing its survival and clonal expansion, and promoting the occurrence of B cell lymphoma. HSC, hematopoietic stem cells; HEV, high endothelial venule; FDC, follicular dendritic cell; EBV, Epstein-Barr virus.

**Figure 3 cancers-14-05699-f003:**
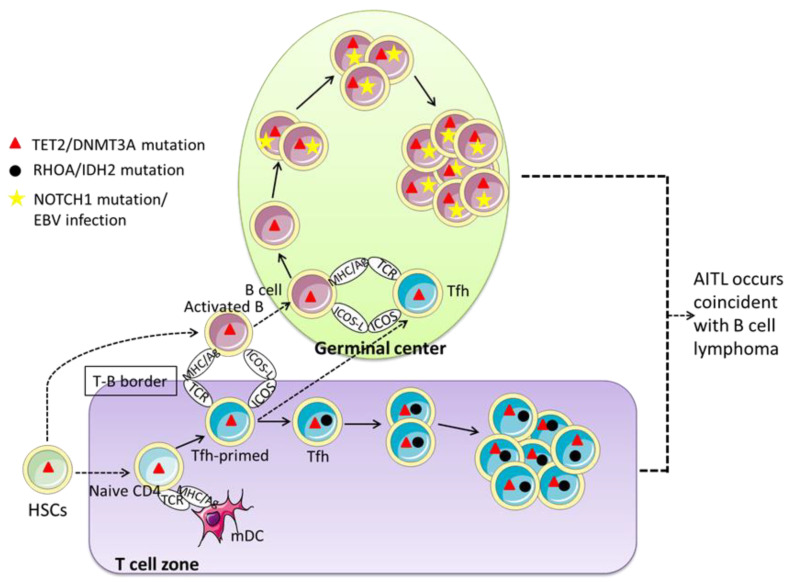
Schematic model of co-occurrence of AITL and B cell lymphoma. This multistep and multilineage model shows that *TET2*/*DNMT3A* mutation in HSCs will be an initiating event in the process of transformation, inducing the generation of multipotent premalignant T and B cell lines. In the T cell zone, naive CD4+ T cells contact with myeloid DC cells and migrate to the T-B border, activating B cells. Upon acquisition of the second hit in *RHOA*/*IDH2* mutation, Tfh cells will be abnormally activated and proliferate, eventually inducing the development of AITL. B cells that already carry the *Tet2*/*DNMT3A* mutation are derived from activated B cells. Correspondingly, subsequent acquisition of *NOTCH1* mutation/EBV infection in B cells may lead to the transformation of B cell lymphoma. Eventually, AITL and B Cell lymphoma develop simultaneously. HSC, hematopoietic stem cells; mDC, myeloid dendritic cell; MHC/Ag, antigen presented on major histocompatibility complex; TCR, T-cell receptor; *ICOS*, inducible T-cell co-stimulator; ICOS-L, ICOS ligand; AITL, angioimmunoblastic T-cell lymphoma.

**Table 1 cancers-14-05699-t001:** Frequencies of *TET2* mutation in AITL.

Diagnosis	Experiment Type	AITL Cases	Mutation Cases	Mutation Rate	Mutation Sites	Mutation Type (Rate)	Amino Acid Change	Mult-Mutation Rate	Reference
AITL& PTCL-NOS	Microarray	30	10	AITL: 33.3%; PTCL-NOS: 20%	NA	NA	NA	NA	Cyril Quivoron [4] et al. (2011)
AITL& PTCL-NOS	direct sequencing	86	40	AITL: 47%; PTCL-NOS: 38%	NA	NA	NA	NA	Lemonnier [18] et al. (2012)
AITL	NGS	85	65	76%	115	Missense: 20/115 (17%); Nonsense: 38/115 (33%); Splice site: 3/115 (3%); Frameshift: 54/115 (47%);	p.Q673*, p.Q727*, p.Q765*, p.R1486*, p.Y1148*	43/65 (66%)	Odejide [30] et al. (2014)
AITL& PTCL-NOS	Targeted resequencing	46	38	AITL: 82.6%; PTCL-NOS: 48.5%	70	Frameshift: 29/70 (42%); Missense: 19/70 (27%); Nonframeshift: 1/70 (1%); Nonsense: 16/70 (23%); Splice site: 5/70 (7%)	p.E1318 splice;	28/38 (74%)	Sakata-Yanagimoto [7] et al. (2014)
AITL& PTCL-NOS	Targeted resequencing	39	32	AITL: 82.1%; PTCL-NOS: 46.3%	48	NA	NA	13/32 (41%)	Wang [24] et al. (2015)
AITL	WES	9	9	100%	15	frameshift or nonsense changes: 14/15 (93%)	NA	6/9 (67%)	Wang [19] et al. (2017)
AITL	Sanger	13	12	92%	15	premature stop codons or deletions:11/15 (73%);replacement:4/15 (27%)	NA	NA	Schwartz [9] et al. (2017)
AITL& Nodal PTCL with TFH phenotype& PTCL-NOS	Targeted resequencing	48	36	AITL:75%; Nodal PTCL with TFH phenotype:100% PTCL-NOS: 55.9%	NA	NA	NA	NA	T B Nguyen [8] et al. (2017)
AITL& PTCL-NOS & PTCL-TFH&FTCL	NA	64	31	AITL: 48%; PTCL-NOS:17% PTCL-TFH: 64% FTCL:75%	NA	NA	NA	NA	Dobay [10] et al. (2017)
AITL	Targeted resequencing	12	12	100%	17	NA	NA	7/12 (58%)	Lemonnier [31] et al. (2018)
AITL& PTCL-NOS	Targeted Exon Sequencing	13	5	AITL:38%; PTCL-NOS:31%	65	NA	R126C, G1869W(2/13); N202K; D302Y; Y620; A893T; W1291;	NA	Fernandez-Pol [32] et al. (2019)
AITL	NGS	44	38	86%	60	NA	NA	23/38 (61%)	Julia Steinhilber [26] et al. (2019)
AITL& PTCL-TFH	Fluidigm Access Array& Illumina MiSeq	94	NA	AITL: 72%; PTCL-TFH: 73%	154	frameshift indels or Nonsense changes: 118 (77%)	NA	57%	Yao [20] et al. (2020)
AITL	Targeted sequencing	10	6	60%	6	NA	R550; Q1274; G422Efs’ 5; L34F; Q909; G422Efs’ 5	0/6	Butzmann [21] et al. (2020)
AITL	Targeted sequencing	5	4	80%	8	Frameshift insertion: 2/8 (25%); Nonsilent: 3/8 (37.5%);Frameshift deletion:3/8 (37.5%)	NA	4/5 (80%)	Tran B. Nguyen [23] et al. (2020)
AITL	NGS	28	25	85%	75	Missense: 26/75 (34.7%);Nosens: 22/75 (29.3%);Frameshift: 22/75 (29.3%);Splice: 4/75 (5.3%);CDS-indel: 1/75 (1.3%)	NA	22/28 (79%)	Ye [22] et al. (2021)
AITL	NGS	44	NA	NA	49	Frameshift: 18/49 (36.7%);Missense: 12/49 (24.5%);Splice: 3/49 (6%);Stop_gained: 12/49 (24.5%);Synonymous: 3/49 (6%);3_prime_UTR: 1/49 (2%)	NA	NA	Marta Rodríguez [27] et al. (2021)
AITL& PTCL-TFH	Targeted resequencing	63	49	AITL: 78%; PTCL-TFH: 58%	NA	NA	NA	28/49 (57%)	Lemonnier [25] et al. (2021)

Abbreviations: AITL, angioimmunoblastic T-cell lymphoma; PTCL-NOS, peripheral T-cell lymphoma, not otherwise specified; FTLC, follicular T-cel1 lymphoma; PTCL-TFH, peripheral T-cell lymphoma with TFH phenotype.

**Table 2 cancers-14-05699-t002:** Treatment outcomes of novel monotherapy and combination therapy for refractory and relapsed T-cell lymphoma including AITL.

5	PTCL Subtype	Design	Primary Endpoint	ORR	CR	PR	Median PFS (Months)	Median OS (Months)	Reference
Romidepsin	PTCL *n* = 130 (PTCL-NOS *n* = 67, AITL *n* = 27)	Phase II, Open-Label	CR/Cru	25%	15%	11%	4	NA	Coiffier [72] et al. (2012)
Belinostat	PTCL *n* = 24 (PTCL-NOS *n* = 13, AITL *n* = 3)	Phase II	ORR	25%	8.30%	16.70%	NA	NA	Foss [73] et al. (2015)
Belinostat	PTCL *n* = 120 (PTCL-NOS *n* = 77, AITL *n* = 22)	Phase II, Open-Label, multicenter	ORR	26%	11%	15%	1.6	7.9	O’Connor [74] et al. (2015)
Chidamide	PTCL *n* = 79 (PTCL-NOS *n* = 27, AITL *n* = 10)	Phase II, Open-Label, multicenter	ORR	28%	9%	14%	2.1	21.4	Shi [75] et al. (2015)
Chidamide	PTCL *n* = 256	Phase II, multicenter	ORR	39.06% (PTCL 37.3% AITL 49.23%)	PTCL 8.73% AITL 9.23%	PTCL 28.57% AITL 40%	4.3	NA	Shi [76] et al. (2017)
Romidepsin + CHOP	PTCL *n* = 37 (PTCL-NOS *n* = 9, AITL *n* = 15)	phase 1b/2	ORR	69%	51%	17%	21.3	NA	Dupuis [77] et al. (2015)
Panobinostat + bortezomib	PTCL *n* = 25 (PTCL-NOS *n* = 9, AITL *n* = 8)	Phase II, Open-Label, multicenter	ORR	43% (PTCL 22% AITL 50%)	21.5% (PTCL 11% AITL 25%)	21.5% (PTCL 11% AITL 25%)	NA	NA	Tan [76] et al. (2015)
Chidamide + chemotherapy	PTCL *n* = 127	Phase II, multicenter	ORR	51.18%	NA	NA	5.4	NA	Shi [76] et al. (2017)
Duvelisib + Romidepsin	T-cell lymphoma *n* = 12	Phase I	ORR	50%	NA	NA	NA	NA	Moskowitz [78] et al. (2017)
Duvelisib + bortezomib	T-cell lymphoma *n* = 17	Phase I	ORR	53%	20%	23%	NA	NA	Moskowitz [78] et al. (2017)
5-Azacytidine	AITL *n* = 12 PTCL *n* = 37	Clinic trial	ORR	75%	50%	25%	15	21	Lemonnier [31] et al. (2018)
Duvelisib + Romidepsin	T-cell lymphoma *n* = 39 (PTCL *n* = 22)	Phase I, Parallel Multicenter	ORR	51% (PTCL 55%)	17% (PTCL 27%)	34%	8.8 (PTCL)	NA	Horwitz [41] et al. (2018)
Romidepsin + CHOP	PTCL *n* = 421 (Ro-CHOP *n* = 211)	Phase III	PFS	63%	41%	22%	12	51.8	Bachy [79] et al. (2021)
5-Azacytidine + romidepsin	PTCL *n* = 25 (PTCL-NOS *n* = 4, AITL *n* = 14)	Phase II, multicenter	ORR	61%	48%	13%	8	Not reached	Falchi [80] et al. (2021)
5-Azacytidine + Chidamide	PTCL *n* = 24 (PTCL-NOS *n* = 4, AITL *n* = 15)	Phase II	ORR	68.8% (AITL 72.7%)	31.2% (AITL 36.4%)	37.5% (AITL 36.4%)	NA	NA	Ding [81] et al. (2021)

Abbreviation: PTCL, peripheral T-cell lymphoma; PTCL-NOS, PTCL not otherwise specified; AITL, angioimmunoblastic T-cell lymphoma; CHOP, cyclophosphamide, doxorubicin, vincristine, prednisone; ORR, overall response rate; CR, complete response; Cru, complete response unconfirmed; PR, partial response; PFS, progression-free survival; OS, overall survival; NA, not analyzed.

**Table 3 cancers-14-05699-t003:** Preclinical trials of *TET2* targeting agents.

Drugs	Disease of Study	Models	Mechanism	Limitations	Reference
Bobcat339 (TET enzyme inhibitors)	NA	HT-22 cells	Reduce DNA 5hmC levels in hippocampal	No testing in animal model and clinical trial	Gabriella [94] et al. (2019)
TET-specific inhibitors (TETi76)	MDS	Cell-permeable diethyl ester of TETi76 and different human leukemia cell lines (K562, MEG-01, SIG-M5, OCI-AML5, and MOLM13)	Decrease cytosine hydroxymethylation and restrict clonal out-growth of *TET2* mutant	Potential to replicate the *TET2* mutation	Guan [95] et al. (2020)
Ascorbic acid (AA)	DLBCL	Lymphoma cell lines LY-1 (DLBCL), Karpas 299 (T-cell NHL), and Jeko (mantle cell NHL)	Enhance TET activity and an increase in the hydroxymethylcytosine fraction; reactivate SMAD1	The target, route of administration, and dose are unclear	N Shenoy [96] et al. (2017)
Ascorbate	AML	SKM-1 cells	Increase TET activity	The target, route of administration, and dose are unclear	Carlos [97] et al. (2021)
Ascorbic acid (AA)	Myeloid neoplasia (MN)	*TET2*^−/−^ mice	facilitate Fe(III)/Fe(II) redox reaction	The target, route of administration, and dose are unclear	Guan [98] et al. (2020)
Ascorbic acid (AA) + 5-Azacytidine (5-aza)	Pediatric T-cell acute lymphoblastic leukemia (T-ALL)	*TET2*-silenced T-ALL cells	Stable re-expression of the *TET2* gene; up-regulation of methylated genes and human endogenous retroviruses (HERVs)	The target, route of administration, and dose are unclear	Maike [99] et al. (2021)
Vitamin C	Colorectal cancer (CRC)	HCT 116 cells	Increase expression of *CDKN1A*	The target, route of administration, and dose are unclear	Christian [100] et al. (2018)
Four-antibiotic cocktail	CMML	*TET2* KO mice	Supress TNF-α signaling	How antibiotics inhibit the pathways associated with TNF is unclear	Zeng [101] et al. (2019)
Ginkgo biloba extract (GBE)	Colorectal cancer (CRC)	SW480 cells	Reduce expression of miR-29a	No testing in the clinical trial	Li [102] et al. (2022)

Abbreviation: DLBCL, Diffuse large B-cell lymphoma; CMML, chronic myelocytic leukemia; MDS, myelodysplastic syndromes; AML, acute myelocytic leukemia; NA, Not available; TET, ten-eleven translocation protein; *CDKN1A*, tumor suppressor p21; 5hmc, 5-Hydroxymethylcytosine.

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
