# Peer review of "Targeting *TET2* as a Therapeutic Approach for Angioimmunoblastic T Cell Lymphoma"

_cancers, 2022, doi:10.3390/cancers14225699_

Round 1

Reviewer 1 Report

Dear Editor 

The review by Hu et al. gives a good overview of the role of Tet2 in AITL pathogenesis and the synergistic effects of TET2 mutations and additional genes such as RHOA, DNMT3A and IDH2 revealed in AITL. 

It also reports on the already used drugs interfering with Tet2 mutations effect and reports on new therapeutic possibilities.

This is a clear, comprehensive overview focusing on the role of TET2 in AITL disease. 

Comments

1)    The figures are informative but the legend abbreviations are not put in the legends of the figures but below that as main text. This should be corrected

2)    The Tet2/RhoA combination of mutations in AITL and the characteristics of the mouse models harboring these mutations are described in a separate review, which might be mentioned in this review pint 4.1: 

Mhaidly R, Krug A, Gaulard P, Lemonnier F, Ricci JE, Verhoeyen E. New preclinical models for angioimmunoblastic T-cell lymphoma: filling the GAP. Oncogenesis. 2020 Aug 14;9(8):73. doi: 10.1038/s41389-020-00259-x. PMID: 32796826; PMCID: PMC7427806.

3)    Same remark for the genes implication in epigenetic modifications; a recent review might be cited in this context in point 6:

Tari G, Lemonnier F, Morschhauser F. Epigenetic focus on angioimmunoblastic T-cell lymphoma: pathogenesis and treatment. Curr Opin Oncol. 2021 Sep 1;33(5):400-405. doi: 10.1097/CCO.0000000000000773. PMID: 34230442.

4)    Finally, the author might also refer to a more general review including other treatment options not related to tet2 at the end of this review.

Krug A, Tari G, Saidane A, Gaulard P, Ricci JE, Lemonnier F, Verhoeyen E. Novel T Follicular Helper-like T-Cell Lymphoma Therapies: From Preclinical Evaluation to Clinical Reality. Cancers (Basel). 2022 May 12;14(10):2392. doi: 10.3390/cancers14102392. PMID: 35625998; PMCID: PMC9139536.

Author Response

Reviewer #1: The review by Hu et al. gives a good overview of the role of Tet2 in AITL pathogenesis and the synergistic effects of TET2 mutations and additional genes such as RHOA, DNMT3A and IDH2 revealed in AITL. It also reports on the already used drugs interfering with Tet2 mutations effect and reports on new therapeutic possibilities. This is a clear, comprehensive overview focusing on the role of TET2 in AITL disease.

Minor point:

1) The figures are informative but the legend abbreviations are not put in the legends of the figures but below that as main text. This should be corrected

Answer: Thank you very much for your comments and reminders, corrections have been made and highlighted in red in the revised version.

2) The Tet2/RhoA combination of mutations in AITL and the characteristics of the mouse models harboring these mutations are described in a separate review, which might be mentioned in this review pint 4.1:

Mhaidly R, Krug A, Gaulard P, Lemonnier F, Ricci JE, Verhoeyen E. New preclinical models for angioimmunoblastic T-cell lymphoma: filling the GAP. Oncogenesis. 2020 Aug 14;9(8):73. doi: 10.1038/s41389-020-00259-x. PMID: 32796826; PMCID: PMC7427806.

Answer: We have read this review in detail and found that it comprehensively summarizes the characteristics of the mouse model carrying the combined TET2/RHOA mutation in AITL, which indeed has enriched the content of our paper. We have added a citation to this article at position 48 of the reference.

3) Same remark for the genes implication in epigenetic modifications; a recent review might be cited in this context in point 6:

Tari G, Lemonnier F, Morschhauser F. Epigenetic focus on angioimmunoblastic T-cell lymphoma: pathogenesis and treatment. Curr Opin Oncol. 2021 Sep 1;33(5):400-405. doi: 10.1097/CCO.0000000000000773. PMID: 34230442.

Answer: Thanks for sharing, we have read this review and cited it in reference 75.

4) Finally, the author might also refer to a more general review including other treatment options not related to tet2 at the end of this review.

Krug A, Tari G, Saidane A, Gaulard P, Ricci JE, Lemonnier F, Verhoeyen E. Novel T Follicular Helper-like T-Cell Lymphoma Therapies: From Preclinical Evaluation to Clinical Reality. Cancers (Basel). 2022 May 12;14(10):2392. doi: 10.3390/cancers14102392. PMID: 35625998; PMCID: PMC9139536.

Answer: Thanks for sharing these excellent articles. In fact, this review was read by us before. It summarizes the new therapeutic agents for Follicular Helper-like T-Cell Lymphoma (specifically for AITL and PTCL), which includes not only epigenetic agents but also summarizes the possibilities of targeting various signaling pathways, making it a very comprehensive review. We have cited it as the 72nd reference.

Reviewer 2 Report

It their review, Hu et al comprehensively summarized the current knowledge on the role of synergistic mutations of TET2 and RHOA, DNMT3A, and IDH2 in the pathogenesis of AITL. They also reported the current and future attempts to therapeutically target TET2 in angioimmunoblastic T cell lymphoma. The paper is well written, and there are only minor points to be corrected.

Minor points

All gene abbreviations should be written in italic.

110 “number-neutral loss-of-heterozygosity” should read “copy number-neutral loss-of-heterozygosity”

116 “PTCL, NOS” should read “PTCL-NOS”

126 “hrombocytopenia” should read “thrombocytopenia”

145, 192, 209, 231 “Pathogenesis of interaction” can read just “interaction”.

451 “particularly HDACs” should read “particularly HDAC inhibitors”

453 “the survival rate of TET2 mutant malignant tumors.” should read “the survival rate of patients with TET2 mutant malignant tumors.”

Author Response

Reviewer #2: It their review, Hu et al comprehensively summarized the current knowledge on the role of synergistic mutations of TET2 and RHOA, DNMT3A, and IDH2 in the pathogenesis of AITL. They also reported the current and future attempts to therapeutically target TET2 in angioimmunoblastic T cell lymphoma. The paper is well written, and there are only minor points to be corrected.

Minor point:

1) All gene abbreviations should be written in italic.

Answer: Thank you very much for your comments and reminders, corrections have been made in the revised version and are highlighted in red.

2) 110 “number-neutral loss-of-heterozygosity” should read “copy number-neutral loss-of-heterozygosity”

Answer: Thank you for your assisted revision, the correction has been reflected in the revised version.

3) 116 “PTCL, NOS” should read “PTCL-NOS”

Answer: Thank you very much for your reminders, corrections have been made in the revised version and are highlighted in red.

4) 126 “hrombocytopenia” should read “thrombocytopenia”

Answer: Thank you very much for your reminders, correction has been made in the revised version and is highlighted in red.

5) 145, 192, 209, 231 “Pathogenesis of interaction” can read just “interaction”.

Answer: Thank you for the authentic and concise expressions, which really make our language more accurate.

6) 451 “particularly HDACs” should read “particularly HDAC inhibitors”

Answer: We appreciate your reminder, and the correction has been made in the revised version and is highlighted in red.

7) 453 “the survival rate of TET2 mutant malignant tumors.” should read “the survival rate of patients with TET2 mutant malignant tumors.”

Answer: We appreciate your reminder and careful reading. The changes you suggested all made the language more fluent, and we have made corrections in the revised version individually.